# Effects of Alberta Family Integrated Care (FICare) on Preterm Infant Development: Two Studies at 2 Months and between 6 and 24 Months Corrected Age

**DOI:** 10.3390/jcm11061684

**Published:** 2022-03-18

**Authors:** Amanda M. Moe, Jana Kurilova, Arfan R. Afzal, Karen M. Benzies

**Affiliations:** 1Faculty of Nursing, University of Calgary, Calgary, AB T2N 1N4, Canada; jkurilov@ucalgary.ca (J.K.); arfan.afzal@ucalgary.ca (A.R.A.); benzies@ucalgary.ca (K.M.B.); 2Departments of Community Health Sciences and Pediatrics, Cumming School of Medicine, University of Calgary, Calgary, AB T2N 4N1, Canada

**Keywords:** premature infant, parent–child interactions, mothers, child development, cluster randomised controlled trial, family integrated care

## Abstract

Preterm infants are at increased risk for developmental delays. Family integrated care (FICare) is a novel care delivery model that integrates parents into their infant’s care in the neonatal intensive care unit. Two follow-up studies are presented to identify effects of Alberta FICare™ on the development of preterm infants born between 32 and 34 weeks of gestation. Data for Study 1 were collected at an age of 2 months, and between 6 and 24 months for Study 2. In Study 1, Ages and Stages Questionnaires (ASQ) and maternal psychosocial distress measures were completed by 330 mothers of 387 infants (FICare, *n* = 223; standard care, *n* = 164). Study 2 utilised an additional measure, the Parent–Child Interaction Teaching Scale, with 50 mothers of 61 infants (FICare, *n* = 30; standard care, *n* = 31). For Study 1, there was no effect of Alberta FICare™ on the ASQ domains of communication, problem solving, or personal–social at an age of 2 months. For Study 2, the risk of communication delay was significantly lower for infants in Alberta FICare™ compared with standard care. Results from Study 2 suggest a possible protective effect of Alberta FICare™ for the risk of communication delays between 6 and 24 months. Further investigation into the effect of Alberta FICare™ on parent–child interactions and implications for long-term development is warranted.

## 1. Introduction

In Canada, 8.1% of infants are born preterm at less than 37 weeks of gestation [1]. The 2019–2020 rate of preterm birth in Alberta was 8.8%, which continues to be higher than the national average [1]. The short-term and long-term morbidity of infants born at less than 32 weeks of gestation has been well established [2,3]. Research for infants born between 32 and 36 weeks of gestation (moderate to late preterm) is only beginning to uncover the unique needs and outcomes of this population. Compared with their full-term counterparts, moderate and late preterm infants are at increased risks of developmental delay [4,5,6,7,8,9], cognitive delay [2,4,9,10], communication impairments [7,9,10,11,12], and behaviour challenges [2,5,9]. The risk of these conditions decreases as gestation increases [2,4,7,8]. Two systematic reviews of early childhood neurodevelopment in late preterm infants identified that although this population does not tend to have profound disability, the possibility of modest challenges in more than one developmental domain may compound and culminate as difficulties in a child’s school performance [4,13]. Challenges with cognitive, social, or emotional development in early childhood may lead to struggles later in life. Understanding which factors have the greatest impact on neurodevelopment is crucial to supporting moderate and late preterm infants and their families to improve outcomes.

Neonatal intensive care is a reality for most moderate and late preterm infants and poses challenges for parent–infant relationships. Physical separation, preterm infant characteristics (e.g., irritability and sleepiness), and altered parental roles disrupt early parent–infant relationships [14]. Mothers have reported increased depression, anxiety, and stress when their newborn requires medical care [15,16]. Parents are central figures in their child’s health and development; thus, it is imperative to involve them in the care of their infant(s) early on. Alberta Family Integrated Care (FICare) was trialled as a novel care delivery model in Level II Neonatal Intensive Care Units (NICUs) that care for neonates born ≥32 weeks of gestation [17].

Alberta FICare™ (Calgary, AB, Canada) focuses on empowering parents to build their knowledge, skill, and confidence in caring for their preterm infant while in the NICU. Parents are supported to provide non-medical care, emphasising relational communication, parent education, and parent support. The integration of families into the care of their infants significantly decreased infant length of stay by 2.55 days and did not increase the number of visits to the emergency room or the number of hospital readmissions [17]. However, the extent to which this model of care affects longer-term infant development and parent–child interactions has not been explored with moderate and late preterm infants. In a study with infants ≤ 33 weeks of gestation, investigators found positive effects of FICare on parent-reported infant dysregulation and parent–infant interactions [18]. Using path models, FICare had a direct effect on infant dysregulation; FICare had an indirect effect on infant dysregulation mediated by parent stress related to the child [18]. The majority of preterm infants are born moderate or late preterm [19]; therefore, it is important to ascertain whether Alberta FICare™ has similar positive effects.

### 1.1. Theoretical Framework

Bronfenbrenner’s [20,21] bioecological model is a widely applied theoretical framework for exploring influences on human development. This model asserts that multifactorial interactions between individuals and systems affect a child’s intellectual, emotional, social, and moral development. The systems presented in Bronfenbrenner’s theory involve interactions between (a) the microsystem, which can include individual relationships or experiences with others, such as family members; (b) the mesosystem, which links two or more microsystems, such as transitions between home and health care settings; (c) the exosystem, whereby events or processes affect an individual, but do not directly involve them, such as the relationship between a parent’s workplace and home; and (d) the macrosystem, which encompasses the effects of culture on the characteristics embedded within each of the previous systems [20,21].

Optimal child development depends on the early parent–infant relationship [22]. This relationship is disrupted when an infant requires care in an NICU [23]. Given the higher risk of developmental delays for moderate and late preterm infants, it is particularly important to establish early positive parent–infant interactions. Barnard’s Parent–Child Interaction Model [24,25] introduced a mid-range theory whereby an interdependent relationship exists between caregivers, infants, and the environment. Both caregivers and infants are responsible for appropriately providing and responding to cues within a given environment. Such interactions can be measured on parent and infant domains, which can, in turn, be utilized to identify areas of concern or increased needs for intervention. This framework asserts that the dynamic relationship between parent and child directly impacts long-term infant development.

### 1.2. Objectives

The objectives of our two follow-up studies to the primary Alberta FICare™ study [17] were to identify the effects of Alberta FICare™ on communication, problem solving, and personal–social development of moderate and late preterm infants at 2 months and 6 to 24 months corrected age (CA). We hypothesized that Alberta FICare™ would improve infant development, as measured by the Ages and Stages Questionnaires, Third Edition (ASQ-3) [26], when compared with standard care. In addition, maternal stress, depression, anxiety, and parent–child interaction were investigated as potential covariates to gain further understanding of the longer-term effects of Alberta FICare™ on infant development.

## 2. Methods

### 2.1. Setting

Alberta has a single, publicly funded health care system that serves a population of 4.4 million [27], with approximately 50,000 births per year [28]. Although family-centred care is the currently accepted philosophy of care [29], none of the NICU sites in the original Alberta FICare™ study [17] had previously implemented the fundamental principles of Alberta FICare™. Hospital administrators endorsed and facilitated unit-level changes at intervention sites [17].

### 2.2. Design

The two studies reported here were follow-ups to a cluster randomized controlled trial (cRCT) evaluating the effects of Alberta FICare™ in 10 level II NICUs [17]. Entire NICUs were randomized into five intervention sites and five control sites.

Study 1. The follow-up at 2 months CA included mothers and infants from all 10 Level II NICUs. Trial Registration: clinicaltrials.gov Identifier: NCT02879799.

Study 2. The follow-up between 6 and 24 months CA required direct observation of parent–child interactions; therefore, only mothers whose infants were cared for in one of four NICUs (two Alberta FICare™ and two standard care) within 1.5 hours’ drive of the university were included. The infant age range for data collection was larger than anticipated due to challenges in contacting mothers of preterm infants and scheduling in-home visits. Trial Registration: clinicaltrials.gov Identifier: NCT03357458.

### 2.3. Samples

Participants in the primary cRCT were recruited between December 2015 and July 2018 [17]. Mothers and their preterm infants born between 32^0/7^ weeks and 34^6/7^ weeks gestational age with a primary admission, or transfer within 72 h, to one of the Level II NICUs were eligible to participate. Infants born at this gestation were included to ensure a minimum exposure of one week to the intervention because they are typically discharged at 36^0/7^ weeks, if medically indicated. Mothers with serious social or health problems, those who birthed triplets or higher-order multiples, or those who could not communicate in English were excluded, as were infants with severe congenital or chromosomal anomalies or who required palliative care. Totals of 654 mothers and 765 infants were enrolled in the cRCT.

Study 1. Data for follow-up Study 1 were collected between 5 July 2016 and 31 October 2018. A total of 359 mothers of 412 infants responded to the 2-month survey: a response rate of 58.5% (66.9% for the FICare group, 50.0% for the standard care group, *p* < 0.001). Mothers who participated in the 2-month follow-up were significantly more likely to be partnered (96.6% versus 90.0%), Caucasian (74.9% versus 58.1%), born in Canada (81.0% versus 69.5%), have a college or university degree (58.5% versus 45.3%), and report an annual family income of over CAD 80,000 (64.4% versus 42.9%) than mothers who did not participate (all *p*-values ≤ 0.001). The surveys of 29 mothers were excluded; 24 surveys were not completed within the time frame required for the ASQ-3, and five surveys were discontinued before completing the ASQ-3. The final sample consisted of 330 mothers and 387 infants.

Study 2. Data for follow-up Study 2 were collected between 11 February, 2018, and 3 March 2019. Of the 171 mothers eligible to participate, 55 enrolled in the follow-up study: a response rate of 32.3%. There were no significant differences in sociodemographic characteristics reported at admission to the NICU between mothers who participated in this study and those who did not. One mother withdrew from the study after the home visit, and an additional four were excluded (one did not complete the survey, whereas three discontinued the survey before completing the ASQ-3). The final sample consisted of 50 mothers and 61 infants.

### 2.4. Intervention

Alberta FICare™ is a theoretically driven, psychoeducational model of care that empowers parents to build their knowledge, skill, and confidence to care for their infant(s) in the NICU. It recognises the consequences of the early integration of families into the care of their infant throughout hospitalisation in the NICU to improve overall child development. The model has three components: (a) Relational Communication, based on family systems theory [30]; (b) Parent Education, based on adult learning [31] and self-efficacy [32] theories; and (c) Parent Support, based on stress and coping theory [33]. Various strategies within the Alberta FICare™ intervention could impact longer-term child development. Parents are considered partners with the health care team and are encouraged to participate in decision-making and infant care. Spending time with their infant provides opportunities to interpret and respond to infant cues, and education and support from health care providers. As such, parents are encouraged to be in the NICU for a minimum of six hours each day to attain the greatest benefit from the three Alberta FICare™ components. A detailed description of the intervention can be found elsewhere [17]. Mothers in the standard care group received care as usual.

### 2.5. Measurement

Table 1 shows the measures used in Study 1 and Study 2. The primary outcome was the risk of developmental delay. We only included communication, problem solving, and personal–social domains on the ASQ-3 because we did not expect Alberta FICare™ to affect motor development.

### 2.6. Procedures

Research nurses obtained informed consent from mothers at enrolment to the primary cRCT [17]. Mothers completed electronic surveys through the Qualtrics online survey platform (Qualtrics, Provo, UT, USA) on a tablet at admission and discharge; research nurses extracted infant data from the medical records post-discharge. For follow-up Studies 1 and 2, we contacted mothers to determine interest and sent a link for an electronic survey. Only mothers who agreed to be contacted about future studies were contacted. In addition to the follow-up survey, for Study 2, we conducted a single home visit to observe parent–child interactions when the infant was at least 6 months CA. During a structured play session, specially trained nurses, blind to the study group, digitally video-recorded parent–child interactions. Coders blind to the study group later coded the videos according to the Parent–Child Interaction Teaching Scale (PCITS) [38].

### 2.7. Statistical Analysis

Data obtained from maternal NICU admission and discharge surveys were examined for missing values and patterns of missing values. For scales, missing values were replaced as recommended by developers. Study 1 and Study 2 data were analyzed using 2-level hierarchical logistic regression models for each included ASQ-3 domain. Level-1 was child level variables such as birth weight, which were nested within level-2 groups and shared the impact of level-2 variables in common (i.e., twin children will have the same data for mother variables). Level-2 was mother-level variables such as mother’s age, income, etc. The likelihood ratio test comparing the logistic regression versus hierarchical models was significant for Study 1 models (*p* < 0.05), indicating that a clustering effect was present and that the hierarchical model was preferred. The intraclass correlation coefficients for communication, problem solving, and personal–social models were 0.99, 0.99 and 0.54, respectively, indicating high similarities among the twin observations. The likelihood ratio test comparing the logistic regression versus hierarchical models was not significant for Study 2 models (*p* > 0.05), indicating that no clustering was present. However, to keep the modelling approach consistent, and since most of the Study 2 participants also participated in Study 1, we used a 2-level hierarchical logistic regression model for Study 2, which was reduced to an ordinary logistic regression model without the presence of clustering. The Hosmer–Lemeshow goodness-of-fit tests indicated that both models fitted the data well (*p* > 0.05). Before running each regression model (communication, problem solving, and personal–social), bivariate correlations or chi-squared tests were conducted between the dependent and independent variables, as appropriate. Independent variables that had correlations or associations, with a *p*-value of 0.10 or less, were included in the final models. Finally, for both studies, the *p*-value was set to 0.017 to adjust for multiple comparisons for all inferential analyses. IBM SPSS Version 26 and STATA IC Version 14.2 were used for the statistical analyses.

## 3. Results

Table 2 presents the characteristics of mothers and infants. Participant characteristics did not differ between groups, except in Study 1 the Alberta FICare™ group had a significantly higher proportion of mothers who identified as Caucasian (*p* = 0.047), which was also the case in the primary cRCT [17]. In Study 2, infants in the Alberta FICare™ group had a significantly longer NICU length of stay (*p* = 0.048). This longer length of stay can be attributed to one of the two Alberta FICare™ NICUs being in a smaller regional hospital, where discharges were delayed for families whose rural residence may result in delayed hospital access in the event of infant complications, whereas both standard care NICUs were in large urban hospitals.

Scale scores are presented in Table 3. In Study 1, we included the group (Alberta FICare™ or standard care) in all models to investigate whether Alberta FICare™ improved communication, problem solving, or personal–social development. There was no evidence of an association between groups or risk of developmental delay in any domain. In Study 2, we found a significant association between groups and the communication domain, where the risk of delay in communication was significantly lower for infants in the Alberta FICare™ group compared with standard care (*p* = 0.014, 95% CI 0.01–0.62). There was no evidence of an association between groups and the other ASQ domains. Additional details regarding the covariates included in the final models are available in the Appendix A.

## 4. Discussion

We conducted these two longitudinal follow-up studies to investigate the effects of Alberta FICare™ [17] on development in moderate and late preterm infants. In Study 1, we found no effect of Alberta FICare™ on the ASQ domains of communication, problem solving, or personal–social at 2 months CA. No association was observed between any infant, maternal, or NICU variables and risk of developmental delay in Study 1. In Study 2, there was a significant association between Alberta FICare™ and the ASQ communication domain, suggesting a possible protective effect of Alberta FICare™ on infant development between 6 and 24 months CA when compared with standard care. Similar to Study 1, there was no effect of Alberta FICare™ on the problem solving or personal–social ASQ domains, and no association was observed between any infant, maternal, parent–child interaction, or NICU variables and the risk of developmental delay in Study 2.

Existing interventions to integrate parents into the care of their infant in NICU and their effects on risk of developmental delay show varying results [39]. This is partly due to the large variations in (a) the gestational ages at birth included for different studies, (b) the types of interventions being implemented, and (c) the instruments utilized to measure developmental outcomes [39]. The Bayley Scales of Infant and Toddler Development are the most widely used developmental assessment for infants [40]. In one systematic review, however, even the Bayley Scales produced mixed results when assessing developmental and family-centred care interventions [41]. Although the ASQ is primarily used as a screening tool, evidence suggests that it is sensitive to risks of developmental delay in late preterm infants [11] and could detect group differences.

Our findings from Study 1 are consistent with a follow-up cohort study of extremely preterm infants who participated in a Canadian multicenter FICare cRCT trial which found no significant group differences on the Bayley-III cognitive or language domains at an age of 18 months [42]. This study reported higher motor scores on the Bayley-III assay; however, we did not include motor development as an outcome measure because we hypothesised that Alberta FICare™ would not influence motor development. The Australian Baby Triple *p* for Preterm Infants program [43], which uses multilevel parental interventions, and the Family Nurture Intervention [44], which focused on facilitating maternal-infant emotional connection, each reported higher cognitive and motor scores on the Bayley Scales for infants born at ≤34 weeks. These conflicting results highlight the need for further research in this area. Although the ASQ is an appropriate screening tool to identify infants at risk for developmental delay, utilizing a more comprehensive developmental measurement tool, such as the Bayley Scales, may increase the comparability of outcomes with other studies.

The results from Study 2 suggest a possible protective effect of Alberta FICare™ for the risk of communication delays between 6 and 24 months. This is important because one Canadian study comparing late preterm infants admitted to the NICU with those not admitted to the NICU reported that NICU admission significantly increased the risk of communication delays, as measured by the ASQ-3 at an age of 12 months CA [11]. Our results are consistent with an Australian RCT study of preterm infants where mothers that received a modified Mother–Infant Transaction Program intervention reported a significant group difference (favoring higher scores in the intervention group) in communication but not problem solving at an age of 24 months [45]. Our study did not show any effect of gestational age on the risk for developmental delay. However, the risk of developmental delay is inversely related to gestational age [8]. Given these findings and the impact of NICU admissions [11], it may be prudent to further evaluate the effect of Alberta FICare™ on parent–child interactions with larger samples that enable mediation modelling to understand its effect on later development for moderate and late preterm infants. Alternatively, the effect of Alberta FICare™ on risk of developmental delay may be mediated by parenting stress rather than parent–infant interactions [18].

We hypothesised that integrating parents as members of the health care team and increasing opportunities to care for their infant would reduce maternal psychosocial distress and increase confidence in providing care for their infant, with a consequent positive effect on longer-term child development. However, our analysis showed no relationship between maternal characteristics and any infant developmental domain. Despite infants in the Alberta FICare™ group being discharged 2.55 days earlier than infants in the standard care group, maternal psychosocial distress and confidence were not significantly different between groups at discharge [17]. Both groups improved during hospitalization, even though mothers in the Alberta FICare™ group had less time to reduce psychosocial distress and gain confidence.

### Strengths and Limitations

To the best of our knowledge, this is the first study to explore the effects of FICare on the development of moderate and late preterm infants. However, some limitations should be considered. First, both of our studies included high proportions of primiparous women on maternity leave; both factors that may enable mothers to give their preterm infants a longer duration of focused and individualised attention in the NICU. Canada offers a 50-week parental leave policy, increasing the time children spend with maternal caregivers versus institutional care [46]. Therefore, these results may not be generalizable to jurisdictions without similar parental leave policies. Second, Study 2 had a small sample size, which was partially the result of limiting participants to a geographical region within driving distance to collect observational data. A larger sample size would allow for the inclusion of more variables and a more thorough analysis that includes the impact of measurement subscales on specific outcomes. Finally, neither study included motor development outcomes because we hypothesised that Alberta FICare™ would not affect motor development. Given the conflicting results of other studies on this domain [11,47], future research should include all developmental domains.

## 5. Conclusions

Results from these two studies indicate that future research is needed to understand the long-term effects of Alberta FICare™ on infant development. This recommendation is supported by a systematic review of interventions aimed at improving child development outcomes [48]. Although it would be impracticable to recommend practice changes, our findings from Study 2 are promising. Alberta FICare™ training for health care providers provides the strategies and tools to integrate parents in their infant’s care while in the NICU [17]. Integrating parents into the NICU care team reduces psychosocial distress and increases confidence in caring for their preterm infant, which may improve early parent–child relationships and the development of communication skills. This potential mechanism is worth exploring in future research.

## Figures and Tables

**Table 1 jcm-11-01684-t001:** Description of measures.

Measure	Study Using the Measure	Description
**Child Development**		
Ages and Stages Questionnaires, Third Edition (ASQ-3) [26]	Study 1 and Study 2	A series of 21 age-specific (2 to 60 months of age) questionnaires with 30 items per questionnaire to assess risk of developmental delay across five skill domains: (1) communication, (2) gross motor, (3) fine motor, (4) problem solving, and (5) personal-social. Responses are converted to a point value and summed. Each domain contains cut-off scores to indicate appropriate development, monitoring zone, or referral required. Higher scores indicate more optimal child development. Sensitivity (0.86) and specificity (0.85) are high. We collapsed monitoring and referral categories into one risk of developmental delay category.
**Psychosocial Distress**		
Edinburgh Postnatal Depression Scale (EPDS) [34]	Study 1 and Study 2 up to infant age of 12 months	10 items relating to postnatal depression symptoms measured on a 4-point Likert scale. Scores are summed to provide an overall score to identify risk for postnatal depression. Theoretical scores range from 0 to 30, where higher scores indicate greater depressive symptoms. Any positive score for question 10 requires immediate follow-up. Using a score of ≥13 as a cut-off provides a sensitivity of 0.86 and specificity of 0.78, with a positive predictive value of 73%.
left for Epidemiological Studies Depression Scale Revised (CESD-R) [35]	Study 2 after infant age of 12 months	20 items that measure depression on a 4-point Likert scale. There are nine subscales to assess: (1) sadness, (2) loss of interest, (3) appetite, (4) sleep, (5) thinking/concentration, (6) guilt, (7) tired, (8) movement, and (9) suicidal ideation. Item scores are added to calculate total scores where higher scores indicate greater depressive symptoms. Theoretical scores range from 0 to 60. Cronbach’s α = 0.85 to 0.90 and test–retest reliabilities (0.45 to 0.70) are moderate.
State-Trait Anxiety Inventory (STAI) [36]	Study 1 and Study 2	40 items to assess anxiety on a 4-point scale. 20 items measure trait anxiety, and 20 items measure state anxiety. Item scores are added to calculate total scores, where higher scores indicate greater anxiety. Internal consistency (0.86 to 0.95) and test–retest reliabilities (0.73 to 0.86) are high. Only state anxiety was measured as part of the follow-up studies; trait anxiety was measured only at admission to NICU.
Parenting Stress Index, Fourth Edition Short Form (PSI-4-SF) [37]	Study 1 and Study 2	36 items rated on a 5-point Likert scale to capture three domains: parental distress, parent–child dysfunctional interaction, and difficult child. Items are summed to calculate domain scores and a Total Stress score. Theoretical scores range from 12 to 60 for each domain and 36 to 180 for Total Stress. Raw scores are converted to percentiles; scores at or above the 85th percentile are considered clinically significant. Internal consistency reliability coefficients (0.95 for Total Stress and 0.88 to 0.90 for subscales) are high. Test–retest studies were not conducted for this version.
**Parent–Child Interaction**		
Parent–Child Interaction Teaching Scale (PCITS) [38]	Study 2	Measures the presence or absence of dyadic behaviours on four parent and two infant subscales. The parent subscales have 50 items to assess: (1) sensitivity to cues, (2) response to distress, (3) social-emotional growth fostering, and (4) cognitive growth fostering. The infant subscales have 23 items to assess: (1) responsiveness to caregiver and (2) clarity of cues. Theoretical scores for the parent subscales are 0 to 50 and 0 to 23 for the infant subscales. The sum of the scores presents the total score where higher scores indicate more optimal interactions.

Note: Bold used to delineate categories for measurement scales.

**Table 2 jcm-11-01684-t002:** Infant and mother characteristics.

	Study 1		Study 2	
Characteristic	Alberta FICare™	Standard Care	Alberta FICare™	Standard Care
**Infant Characteristics**	***n* = 223**	***n* = 164**	***n* = 30**	***n* = 31**
Singleton (% yes)	163 (73.1)	110 (67.1)	22 (73.3)	17 (54.8)
Gestational age				
32 weeks	53 (23.8)	26 (15.9)	3 (10.0)	3 (9.7)
33 weeks	62 (27.8)	39 (23.8)	11 (36.7)	8 (25.8)
34 weeks	108 (48.4)	99 (60.4)	16 (53.3)	20 (64.5)
Male (% yes)	120 (53.8)	96 (58.5)	21 (70.0)	20 (64.5)
Caesarean delivery (% yes)	106 (47.5)	79 (48.2)	17 (56.7)	15 (48.4)
Birth weight (g), mean (SD)	2141.6 (378.5)	2118.3 (391.1)	2172.6 (331.7)	2066.6 (397.3)
Length of stay (days), mean (SD)	18.4 (8.3)	19.6 (7.8)	19.5 (6.3)	16.5 (5.1)
**Maternal Characteristics**	***n* = 193**	***n* = 137**	***n* = 26**	***n* = 24**
Age (years), mean (SD) ^a^	31.2 ± 5.4	31.8 ± 5.0	31.7 ± 5.3	33.0 ± 3.8
Primiparous (% yes)	111 (57.5)	84 (61.3)	9 (34.6)	10 (41.7)
Relationship status ^b^				
Single	3 (1.6)	6 (4.5)	2 (7.7)	1 (4.3)
Partnered	184 (98.4)	128 (95.5)	24 (92.3)	22 (95.7)
Education ^d^				
High school diploma or less	36 (18.8)	19 (13.9)	4 (15.4)	2 (8.3)
Postsecondary certificate/diploma	46 (24.1)	37 (27.0)	9 (34.6)	7 (29.2)
College or university degree	109 (57.1)	81 (59.1)	13 (50.0)	15 (62.5)
Annual family income (CAD)				
<80,000	40 (27.8)	33 (23.8)	9 (34.6)	6 (25.0)
≥80,000	125 (48.4)	88 (60.4)	13 (50.0)	18 (75.0)
Prefer not to answer/do not know	26 (23.8)	16 (15.9)	4 (15.4)	0 (0.0)
Employment ^c^				
Employed	12 (6.5)	5 (3.8)	14 (53.8)	9 (39.1)
Maternity leave	138 (74.2)	106 (79.7)	7 (26.9)	9 (39.1)
Other	36 (19.4)	22 (16.5)	5 (19.2)	5 (21.7)
Born in Canada (% yes) ^c^	158 (82.7)	109 (79.6)	21 (80.8)	17 (70.8)
Ethnicity (% Caucasian) ^e^	151 (79.5)	95 (69.9)	22 (84.6)	20 (83.3)

Abbreviations: Alberta FICare™, Alberta Family Integrated Care™, SD, standard deviation, CAD, Canadian Dollar. Note: Bold used to delineate infant characteristics and values from maternal characteristics. Values are *n* (%) except where otherwise indicated. ^a^ Study 1: Alberta FICare™ *n* = 192. ^b^ Study 1: Alberta FICare™ *n* = 187, Standard Care *n* = 134; Study 2: Standard Care *n* = 23. ^c^ Study 1: Alberta FICare™ *n* = 191. ^d^ Study 1: Alberta FICare™ *n* = 186, Standard Care *n* = 133; Study 2: Standard Care *n* = 23. ^e^ Study 1: Alberta FICare™ *n* = 190, Standard Care *n* = 136. *n* varies due to missing data.

**Table 3 jcm-11-01684-t003:** Scale scores.

	Study 1				Study 2			
Measure	*n*	Alberta FICare™	*n*	Standard Care	*n*	Alberta FICare™	*n*	Standard Care
**ASQ-3 Domains, *n* (%)**		***n* = 223**		***n* = 164**		***n* = 30**		***n* = 31**
Communication ^a^	221		164		30		31	
No risk		189 (85.5)		131 (79.9)		**28 (93.3)**		**19 (61.3)**
Risk		32 (14.5)		33 (20.1)		**2 (6.7)**		**12 (38.7)**
Problem Solving	220		163		29		30	
No risk		184 (83.6)		140 (85.9)		27 (93.1)		24 (80.0)
Risk		36 (16.4)		23 (14.1)		2 (6.9)		6 (20.0)
Personal–Social	221		163		30		31	
No risk		196 (88.7)		139 (85.3)		26 (86.7)		26 (83.9)
Risk		25 (11.3)		24 (14.7)		4 (13.3)		5 (16.1)
**Maternal Scales, M (SD)**		***n* = 193**		***n* = 137**		***n* = 26**		***n* = 24**
STAI State Anxiety	187	30.17 (9.66)	135	30.45 (8.99)	26	32.12 (10.61)	23	27.17 (7.54)
STAI Trait Anxiety ^b^	192	34.08 (8.28)	137	34.91 (8.47)	26	34.04 (7.69)	24	33.13 (8.46)
PSI-4-SF Total Score	188	62.96 (18.23)	135	66.36 (18.63)	26	60.46 (16.52)	23	59.61 (14.53)
PSI-4-SF Subscales								
Parental Distress	188	23.28 (8.00)	136	24.88 (8.47)	26	23.31 (7.51)	23	22.39 (6.51)
Parent–Child Dysfunctional Interaction	188	18.60 (5.89)	136	19.37 (6.39)	26	16.85 (5.23)	23	17.17 (5.17)
Difficult Child	189	21.04 (6.68)	135	22.31 (6.79)	26	20.31 (5.96)	23	20.04 (5.56)
EPDS	187	5.02 (4.40)	136	5.46 (4.51)		-		-
Risk of Depression (% yes) ^c^		-		-	26	3 (11.5)	23	0 (0.0)
PCITS Total		-		-	24	55.50 (5.41)	24	56.96 (5.30)

Abbreviations: ASQ-3, Ages and Stages Questionnaire, Third Edition; EPDS, Edinburgh Postnatal Depression Scale; STAI, State-Trait Anxiety Inventory; PSI-4-SF, Parenting Stress Index, Fourth Edition Short Form; PCITS, Parent–Child Interaction Teaching Scale. ^a^ In Study 2, frequencies and percentages in bold are statistically significant predictors for group in the hierarchical model for Communication (*p* = 0.014, 95% CI 0.01–0.62). ^b^ Measured only at NICU admission. ^c^ Two different depression measures were used in Study 2 based on child age; therefore, scores were recoded into risk/no risk categories. *n* varies due to missing data.

## Data Availability

Data will be available to qualified researchers upon request to the senior author.

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
