# Peer review of "Effects of Alberta Family Integrated Care (FICare) on Preterm Infant Development: Two Studies at 2 Months and between 6 and 24 Months Corrected Age"

_jcm, 2022, doi:10.3390/jcm11061684_

Round 1

Reviewer 1 Report

Hello authors.  Congratulations on putting forth the effort to studying this important topic.  I have very few questions/suggestions. 

2.2 Design.  Please explain briefly the cRCT.  Did it randomize mothers and patients to standard of care vs the FiCare arm?

2.4 Intervention.  This paragraph seems to talk about the model more than it does to describe an intervention.  Therefore, the reader is not clear on the intervention.  Is the intervention encouraging families to be in the NICU for 6 hours daily?  Please clarify.

Table 1.  spacing seems off some.  Please consider revising some. 

Author Response

Thank you for your detailed comments.

2.2 Design.  Please explain briefly the cRCT.  Did it randomize mothers and patients to standard of care vs the FiCare arm?

2.2 Design: We provided additional information about cluster randomization. This section now reads:

The two studies reported here were follow-ups to a cluster randomized controlled trial (cRCT) evaluating the effects of Alberta FICare in 10 level II NICUs [17]. Entire NICUs were randomized into five intervention sites and five control sites.

2.4 Intervention.  This paragraph seems to talk about the model more than it does to describe an intervention.  Therefore, the reader is not clear on the intervention.  Is the intervention encouraging families to be in the NICU for 6 hours daily?  Please clarify.

2.4 Intervention. The Alberta FICare intervention is comprised of three components: Relational Communication, Parent Education, and Parent Support. We clarified the three components by italicizing “…three components:” and seriating the list of the three components. Families were encouraged to be present in the NICU for a minimum of 6 hours per day or three feedings (assuming q3h feedings for moderate and late preterm infants). Benzies et al. 2020 report that mothers in the Alberta FICare group were present for 9.0 (SD = 5.35) hours per day and mothers in the standard care group were present for 7.79 (SD = 4.87) hours per day. This section now reads:

The model has three components: (a) Relational Communication, based on family systems theory [30]; (b) Parent Education, based on adult learning [31] and self-efficacy [32] theories; and (c)  Parent Support, based on stress and coping theory [33]. Various strategies within the Alberta FICare intervention could impact longer-term child development. Parents are considered partners with the health care team and are encouraged to participate in decision-making and infant care. Spending time with their infant provides opportunities to interpret and respond to infant cues, and education and support from health care providers. As such, parents are encouraged to be in the NICU for a minimum of six hours each day for the greatest benefit of the three Alberta FICare components. A detailed description of the intervention can be found elsewhere [17]. Mothers in the standard care group received care as usual.

Table 1.  spacing seems off some.  Please consider revising some. 

Thank you for noticing the formatting error in Table 1. We have corrected.

Reviewer 2 Report

Dear authors,

This is an interesting manuscript; I appreciate your hard work, honesty of the results interpretation and appreciation of the study limitation. Thus, although the results of the study are important, they cannot be generalized. Continuation of the study with the inclusion of more variables is therefore necessary.  

Author Response

Thank you for your words of encouragement. We sincerely appreciate recognition of the intense work required to conduct research and write a manuscript. We carefully reviewed the Discussion and Limitations sections to ensure that we have not overstated our findings, and that we are clear about the need for future research. The cluster randomized controlled trial reported in in this manuscript has concluded. Therefore, there is no opportunity to collect additional data. We added a phrase to strengthen our recommendation that future research is needed. This sentence now reads:

A larger sample size would allow for the inclusion of more variables and a more thorough analysis that includes the impact of measurement subscales on specific outcomes.